# Bayesian reanalysis of early remdesivir for the treatment of COVID-19 in outpatients with high risk of progression to severe disease

**Mazin Abdelghany**[1]*, **Fang Yu**[2], **Stephen Rennard**[3], **Yeongjin Gwon**[2]

**1** Gilead Sciences, Inc., Foster City, California, United States of America, **2** Department of Biostatistics, University of Nebraska Medical Center, Omaha, Nebraska, United States of America, **3** Division of Pulmonary, Critical Care & Sleep Medicine, University of Nebraska Medical Center, Omaha, Nebraska, United States of America

* mazin.abdelghany@mrc-bsu.cam.ac.uk

## Abstract

### Background

Though Bayesian methods are flexible, intuitive, and readily incorporated into clinical decision-making, with particular utility when prior information is available, they remain underutilized in the analysis of clinical trials.

### Methods

In PINETREE, a Phase 3 randomized controlled trial (RCT) of remdesivir (RDV) for the treatment of outpatients with COVID-19 at high risk of severe disease, the primary outcome of COVID-19–related hospitalization or all-cause death was reanalyzed using a range of reference and data-driven priors. Posterior probability distributions were used to calculate the probability that the estimated hazard ratio (HR) was below a range of clinically meaningful specified thresholds and to estimate the treatment effect and its 95% credible interval (CrI).

### Results

Under a minimally informative prior, the posterior probability of an estimated HR less than 1 for COVID-19–related hospitalization or all-cause death was 1 with a posterior median HR 0.13 and 95% CrI 0.02–0.47, recovering the frequentist estimates. Moreover, estimated posterior probability distributions, posterior median HRs, and 95% CrIs were robust across a range of both reference and data-driven prior choices, indicating the strength of the trial data. Lastly, using priors that incorporate historical RCT data, precision of the estimated posterior median HR and 95% CrI was improved over naïve, frequentist estimates.

**Data availability statement:** A data sharing statement has been included as part of the submission. This statement was posted on December 22, 2021, at NEJM.org and is replicated in this submission. The complete de-identified patient data set collected for this study will be made available to others. Requests are at Gilead's discretion and dependent on the nature of the request, the merit of the research proposed, availability of the data and the intended use of the data. If Gilead agrees to the release of clinical data for research purposes, the requestor will be required to sign a data sharing agreement (DSA) in order to ensure protection of patient confidentiality prior to the release of any data. Upon execution of the DSA, Gilead will provide access to a patient-level clinical trial analysis datasets in a secured analysis environment. Code for the analysis is available for review at https://github.com/mazin-abdelghany/pinetree-bayesian-reanalysis.

**Funding:** The author(s) received no specific funding for this work.

**Competing interests:** MA is a former employee of Gilead Sciences, Inc. YG, SR, and FY have no conflicts of interest to declare. This does not alter our adherence to PLOS ONE policies on sharing data and materials. The PINETREE Phase 3 trial data, which we used for this analysis, was utilized for submission of regulatory updates to VEKLURY (remdesivir) prescribing information globally. VEKLURY (remdesivir) is a globally marketed product.

### Conclusions

In a Bayesian reanalysis of the PINETREE trial, there was a 98.9% or greater probability that treatment with RDV reduced the risk of COVID-19–related hospitalization or all-cause death across all prior probability distributions.

---

## Introduction

Frequentist methods are the predominant statistical tool used to analyze randomized clinical trials (RCTs) [1]. Applied to a single trial at a time, frequentist statistics evaluate study hypotheses indirectly by calculating the probability of observing the treatment effect given that the null hypothesis of no treatment effect is true (called the *p* value) [2,3]. Evidence against the null hypothesis is then used to infer that the alternative hypothesis is true [4]. In contrast, Bayesian methods incorporate beliefs, which can include domain expertise or historical data (e.g., other clinical trials or observational data) into the range of plausible treatment effect values. Representing this plausible range as a probability distribution (the prior distribution), new experimental data is integrated with previous knowledge. This provides an updated range of plausible treatment effect values called the posterior distribution [3,5]. The posterior distribution is then used to test research hypotheses directly, for example, by calculating the posterior probability of having a clinically meaningful treatment effect given the newly observed data [6,7]. This method of reasoning aligns with scientific intuition and yields easy-to-understand probability statements [8].

Bayesian methods are particularly useful when a wealth of prior information can be incorporated to estimate the effect of treatment. During the severe acute respiratory syndrome coronavirus 2 (SARS-CoV-2) pandemic, hundreds of clinical trials assessing the efficacy of treatment for coronavirus disease 2019 (COVID-19) were conducted. A Phase 3 clinical trial in non-hospitalized patients with COVID-19 who were at high risk of severe disease showed that a 3-day course of remdesivir (RDV) resulted in an 87% lower hazard of the primary endpoint of COVID-19–related hospitalization or all-cause death compared to placebo (PINETREE) [9]. Here, we present a reanalysis of this primary endpoint in PINETREE, demonstrating the value of utilizing a Bayesian approach to incorporate available prior study results to improve the precision of treatment effect estimates in the setting of multiple available historical studies. Moreover, by exploring several prior distributions—including both reference priors and data-driven priors that incorporate study information from prior RCTs—we demonstrate the intuitiveness, the flexibility in interpretation providing example of clinical reasoning, and the advantages of applying this approach to the analysis of clinical trials.

## Methods

### PINETREE clinical trial design and statistical methods

RDV is a direct-acting, nucleotide prodrug inhibitor of the SARS-CoV-2 RNA-dependent RNA polymerase with a high barrier to resistance [10,11] approved for the

treatment of COVID-19 in adults and children [12]. PINETREE was a randomized, double-blind, placebo-controlled trial of nonhospitalized patients with COVID-19 with at least one risk factor for progression to severe disease (S1 Appendix) [9]. Patients were randomized 1:1, stratified by residence in a skilled nursing facility (yes or no), age (<60 years or ≥60 years), and country (United States or outside United States), to receive intravenous RDV or placebo. 562 patients were randomized and received at least one dose of either RDV or placebo (Fig 1). The primary efficacy endpoint was proportion of participants with COVID-19–related hospitalization or death from any cause. In the original analysis, hazard ratios, two-sided 95% confidence intervals (CIs), and $p$ values for the primary end point were calculated using a Cox proportional-hazards model adjusted for the stratification factors [13].

## Selection of RCTs for data-driven priors

To leverage the data of the RCTs conducted prior to PINETREE study completion, ClinicalTrials.gov was utilized to search for relevant trials to incorporate into data-driven priors. Adult and older adult interventional, Phase 3 studies of COVID-19 with study start between 1 November 2019 and 31 December 2021 were included in the search. The search stop date was selected to correspond with inclusion of trials that started in the same year that the PINETREE trial completed enrollment to minimize variation in treatment effect estimates from rapidly evolving SARS-CoV-2. 273 trials matched the above

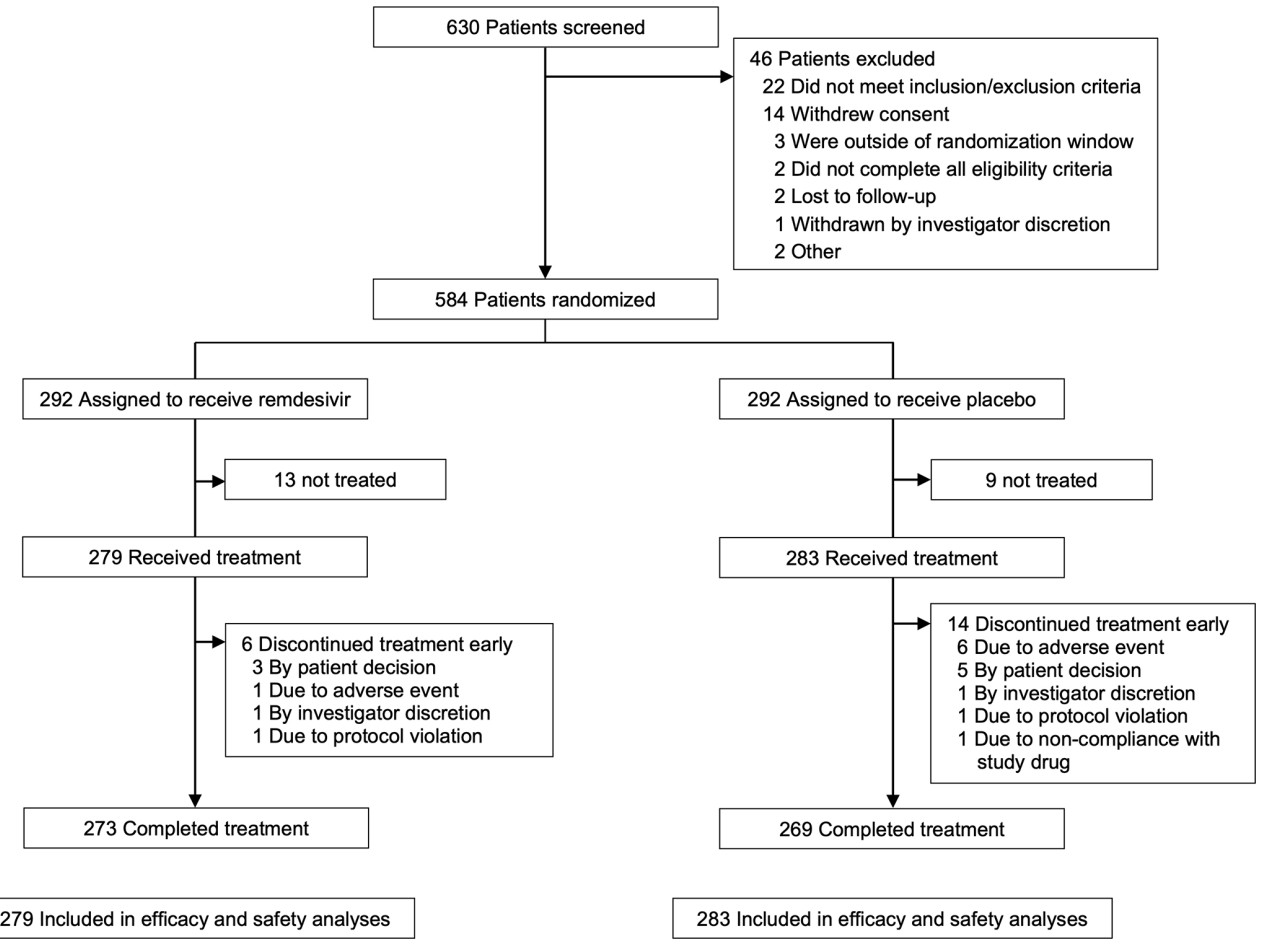

**Fig 1. Patient disposition.**

search criteria. We excluded studies that recruited less than 300 participants, that were designed as prevention trials, that studied non-systemic treatments, and that did not have publicly available results. RCTs with patient populations that overlapped with PINETREE and those with an outcome measure compatible with inclusion in the analysis (i.e., reporting a hazard, risk, or odds ratio [HR, RR, OR]) were included. In total, 9 RCTs (S1 Table) meeting these criteria were selected and used for generation of data-driven priors for the Bayesian reanalysis [14–16].

## Bayesian statistical methodology and support for prior distribution selection

We performed a reanalysis of the prespecified primary end point using a Bayesian proportional hazards model adjusting for the PINETREE stratification factors noted above. Our aim was to estimate the posterior probability distribution of the HR under a range of both reference and data-driven priors, affording us the flexibility to calculate probabilities of clinically important treatment effects (e.g., HR < 1, HR < 0.5, etc.). The posterior probability estimates of the HR that we selected to report in Table 3 incorporated several considerations. First, the convention in frequentist analyses is to set the null hypothesis as a treatment having no benefit (i.e., HR = 1); thus, in our Bayesian reanalysis the probability of any benefit in reduction of COVID-19–related hospitalization or all-cause death was calculated (HR < 1). Second, the 9 RCTs that were selected for inclusion in the data-driven priors had point estimates for the HR of reduction in their primary outcomes ranging from 0.14 to 1.11. Therefore, we reported equally spaced intervals for HR < 1 to HR < 0.1. Because previous trial results of remdesivir showed minimal concern for harm [17,18], posterior probabilities for point-estimates of the HR > 1 were not reported, though can be easily calculated as 1 minus the probability of HR < 1.

In a Bayesian analysis, prior beliefs about the range of plausible treatment effect values are represented by a probability distribution where the variance of the distribution represents the uncertainty about the treatment effect and the area under the curve to the left of a cutoff value equals the probability that the treatment effect parameter (in this case the HR) is smaller than that value.

A range of prior distributions was selected to assess the robustness of the estimated posterior distribution and its resulting conclusions. Two groups of prior distributions were used: (1) reference priors with varying degrees of skepticism of the benefit of RDV for the treatment of COVID-19 and (2) data-driven priors using the 9 RCTs. With evidence gathered on RDV at the beginning of the COVID-19 pandemic noting a trend toward benefit [18], we assessed five reference priors: (1) minimally informative, (2) weakly skeptical, (3) moderately skeptical, (4) weakly pessimistic, and (5) weakly optimistic normal prior distributions on the logarithm of the HR (log(HR); Table 1 and Fig 2A). The strength of the prior belief and the shape of the prior distributions were selected using published recommendations [5]. The minimally informative prior was selected to include HR estimates within a clinically plausible range—HRs from 0.001 to 1000 from the 1st to the 99th percentile of the probability distribution, respectively. Weak and moderate strength of evidence were defined as having standard deviations on the normal distribution of 1 and 0.5, respectively (stronger evidence implies narrower probability distribution, i.e., smaller standard deviation). Skeptical priors were centered on a HR of 1 (no effect), the optimistic prior on 0.75 (25% risk reduction in the primary outcome), and the pessimistic prior on 1.25 (25% risk increase in the primary outcome, i.e., causing harm).

Data-driven priors were constructed using the 9 selected RCTs. For each trial, a normal distribution on the logarithm of the treatment effect estimate (HR, RR, or OR) was generated using the trial's point estimate and CI. The mean of the normal distribution was set to be equal to the logarithm of the trial's point estimate and 95% of the prior probability density of the normal distribution was set over the logarithm of the trial's reported 95% CI (S2 Appendix). Three mixture priors were generated: (1) a mixture distribution of all 9 RCTs, (2) a mixture of RCTs that tested direct-acting antivirals (DAAs), and (3) a mixture of RCTs that tested non-DAAs (Fig 2B). Mixtures were generated by weighting the normal distribution calculated for each trial by a proportion equal to the sample size of each trial divided by the total sample size of all trials included in each mixture (S2 Appendix). Generation of mixture priors were verified using Markov chain Monte Carlo (MCMC) sampling (with 4 chains, 1,000 iterations of warm-up, and 100,000 saved iterations per chain) and used to calculate the prior

**Table 1. Reference prior probability distributions representing prior beliefs about treatment effect of RDV on COVID-19–related hospitalizations or all-cause death.**

| Prior belief | Assumed mean HR | Assumed SD of logarithm of HR | Probability of treatment effect (HR) exceeding specified threshold | | | | | | Rationale for distribution characteristics |
|---|---|---|---|---|---|---|---|---|---|
| | | | > 1.25 | < 1 | < 0.75 | < 0.6 | < 0.4 | < 0.2 | |
| Minimally informative | 1 | 3 | 0.47 | 0.5 | 0.46 | 0.43 | 0.38 | 0.30 | All HR values approx. equally likely; HR values restricted to within clinically plausible range (i.e., 0.99 probability of HR 0.001 to 1000). |
| Weakly skeptical | 1 | 1 | 0.41 | 0.5 | 0.39 | 0.30 | 0.18 | 0.05 | Probability of benefit and harm are equally likely; 0.95 probability of HR between 0.14 and 7. |
| Moderately skeptical | 1 | 0.5 | 0.33 | 0.5 | 0.28 | 0.15 | 0.03 | 0 | Probability of benefit and harm are equally likely; 0.95 probability of HR between 0.38 and 2.7. |
| Weakly pessimistic | 1.25 | 1 | 0.5 | 0.41 | 0.3 | 0.23 | 0.13 | 0.03 | Probability of harm (HR > 1) approx. 0.6; SD near all trials mixture; 0.95 probability of HR between 0.18 and 8.9 represents uncertainty in treatment effect. |
| Weakly optimistic | 0.75 | 1 | 0.3 | 0.61 | 0.50 | 0.41 | 0.26 | 0.09 | Probability of benefit (HR < 1) approx. 0.6; SD near all trials mixture; 0.95 probability of HR between 0.10 and 5.3 represents uncertainty in treatment effect. |

HR = hazard ratio; SD = standard deviation; approx. = approximately

mean, standard deviation, median, and prior probabilities of treatment effects exceeding certain thresholds to compare with the theoretical distributions of the respective mixtures (Table 2).

Separate Bayesian models were generated for each of the 8 prior distributions on the log(HR) of the primary outcome (S2 Appendix). Proportional hazards models were fit by calculating the partial likelihood accounting for the order of the survival times as defined by Breslow [19] for the observed participants (excluding participants whose failure time was censored). MCMC sampling (with 4 parallel chains, 5,000 iterations of warm-up, and 20,000 saved iterations per chain) was used to calculate posterior median estimates of treatment effect and 95% credible intervals (CrIs), and to calculate the posterior probabilities of treatment effects exceeding certain thresholds. Model convergence was assessed visually using trace plots of the generated Markov chains as well as leave-one-out cross-validation (LOO-CV) statistics (including R-hat and effective sample size) [20]. All analyses were conducted in R (R Foundation) version 4.4.2 [21], using RStan package version 2.32.7 [22] to run Stan version 2.32.2 [23]. LOO-CV was performed using the R package loo, version 2.8.0 [24].

## Results

### Original PINETREE trial results

Two of 279 patients (0.7%) in the RDV group and 15 of 283 (5.3%) in the placebo group had a COVID-19–related hospitalization through day 28. No patients in either group died through day 28. RDV was shown to reduce the risk of hospitalization or death through day 28 from any cause by 87% (HR 0.13; 95% CI 0.03–0.59; $p = 0.008$) [9].

### Bayesian reanalysis using reference priors

Using a minimally informative prior, RDV was estimated to reduce the risk of hospitalization or all-cause death through day 28 by 87% (HR 0.13; 95% CrI 0.02–0.47), recapitulating the results of the frequentist analysis. Under this prior, the posterior probability of any reduction in hospitalization or all-cause death with RDV (i.e., probability of HR < 1) was 1, and the probability of HR less than 0.2 was 0.72 (Table 3). Across all reference priors assessed, the posterior probability of any reduction in hospitalization or all-cause death with RDV versus placebo was 0.989 or greater (Table 3 and Fig 3). The

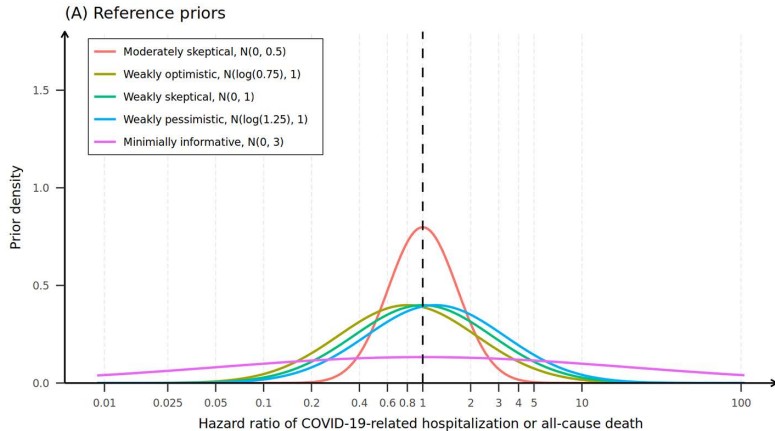

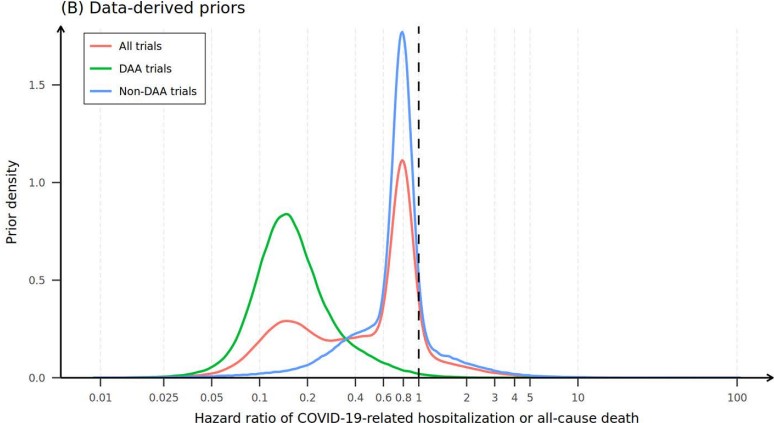

**Fig 2. Reference and data-driven prior distributions for the HR of the treatment effect of RDV on COVID-19–related hospitalization or all-cause death.** HR = hazard ratio; RDV = remdesivir; DAA = direct-acting antiviral. The vertical dotted reference line indicates the no treatment effect threshold (i.e., HR = 1). The distributions in the legend (e.g., N(0, 3)) indicate a normal distribution on the logarithm of the hazard ratio with the x-axis labels exponentiated for convenience. **(A)** Five reference priors were selected to represent a range of beliefs of the treatment effect of RDV on COVID-19–related hospitalization or all-cause death (see Table 1 for rationale for each distribution). **(B)** Three data-driven priors based on 9 previous RCTs (see Methods for details).

**Table 2. Data-driven prior probability distributions representing data collected in other RCTs on treatment effect of therapies for COVID-19–related hospitalizations or all-cause death.**

| Prior belief | Mean HR | SD of logarithm of HR | Probability of treatment effect (HR) exceeding specified threshold | | | | | |
|---|---|---|---|---|---|---|---|---|
| | | | > 1.25 | < 1 | < 0.75 | < 0.6 | < 0.4 | < 0.2 |
| Mixture of all prior trial data | 0.43 | 0.91 | 0.06 | 0.9 | 0.64 | 0.48 | 0.39 | 0.25 |
| Mixture of DAA trial data | 0.16 | 0.58 | 0.002 | 0.994 | 0.985 | 0.97 | 0.93 | 0.70 |
| Mixture of non-DAA trial data | 0.70[a] | 0.62 | 0.09 | 0.86 | 0.48 | 0.25 | 0.14 | 0.04 |

HR = hazard ratio; SD = standard deviation; DAA = direct-acting antiviral

a Median HR = 0.76. Mean and median HR for DAA trials mixture and all trials mixture were nearly equal.

Table 3. Posterior probability of treatment effects estimated by Bayesian analysis by prior beliefs about treatment effect of RDV on COVID-19–related hospitalizations or all-cause death.

| Prior belief | Posterior median HR (95% CrI[a]) | Probability of treatment effect (HR) exceeding specified threshold, % | | | | | | | | | |
|---|---|---|---|---|---|---|---|---|---|---|---|
| | | < 1 | < 0.9 | < 0.8 | < 0.7 | < 0.6 | < 0.5 | < 0.4 | < 0.3 | < 0.2 | < 0.1 |
| **Reference priors** | | | | | | | | | | | |
| Minimally informative | 0.13 (0.02–0.46) | 1 | 0.999 | 0.998 | 0.997 | 0.992 | 0.982 | 0.956 | 0.89 | 0.72 | 0.36 |
| Weakly skeptical | 0.24 (0.08–0.63) | 0.999 | 0.997 | 0.993 | 0.985 | 0.968 | 0.93 | 0.84 | 0.67 | 0.37 | 0.06 |
| Moderately skeptical | 0.44 (0.21–0.89) | 0.989 | 0.977 | 0.952 | 0.90 | 0.80 | 0.63 | 0.39 | 0.14 | 0.02 | 0 |
| Weakly pessimistic | 0.25 (0.08–0.66) | 0.998 | 0.996 | 0.991 | 0.981 | 0.96 | 0.915 | 0.82 | 0.63 | 0.33 | 0.05 |
| Weakly optimistic | 0.22 (0.07–0.59) | 0.999 | 0.998 | 0.995 | 0.99 | 0.977 | 0.947 | 0.88 | 0.72 | 0.43 | 0.08 |
| **Data-driven priors** | | | | | | | | | | | |
| Mixture of all prior trial data | 0.15 (0.06–0.60) | 1 | 0.999 | 0.995 | 0.985 | 0.975 | 0.964 | 0.94 | 0.88 | 0.70 | 0.15 |
| Mixture of DAA trial data | 0.14 (0.06–0.33) | 1 | 1 | 1 | 1 | 0.999 | 0.997 | 0.991 | 0.96 | 0.81 | 0.18 |
| Mixture of non-DAA trial data | 0.27 (0.06–0.79) | 0.999 | 0.994 | 0.977 | 0.938 | 0.89 | 0.85 | 0.76 | 0.57 | 0.30 | 0.08 |

[a] equal-tail interval

CrI = credible interval; HR = hazard ratio

estimated HR reduction in hospitalization or death through day 28 with RDV was weakest using the moderately skeptical prior, 0.44 (95% CrI 0.21–0.89), and the posterior probability of HR less than 0.2 was 0.02 (Table 3).

## Bayesian reanalysis using data-driven priors

The posterior probability of any reduction in hospitalization or all-cause death with RDV versus placebo, HR < 1, was 0.999 or greater across all data-driven priors, and the probability of HR < 0.6 was approximately 0.90 for all 3 data-driven priors, indicating that there is a 0.90 probability that remdesivir reduces hospitalization or all-cause death by 40% regardless of data-driven prior used (Table 3 and Fig 4). Using the data-driven mixture prior of all 9 RCTs, RDV was estimated to reduce hospitalization or all-cause death by day 28 by 85% (posterior median HR 0.15; 95% CrI 0.06–0.60), and the probability of HR less than 0.2 was 0.7. The data-driven mixture prior of the DAA RCTs and the non-DAA RCTs yielded posterior median HRs of 0.14 (95% CrI 0.06–0.33) and 0.27 (95% CrI 0.06–0.79), respectively.

The data-driven mixture prior of all 9 RCTs is bimodal with a mode at HRs between 0.5 and 1 representing the non-DAA trials and a second mode at HRs between 0.1 and 0.2 representing the DAA trials (Fig 4). Notably, when the likelihood (i.e., data) from PINETREE is combined with this prior distribution, most of the probability mass is reallocated to the second mode as the data are most consistent with those of the DAA trials in the mixture prior. In contrast, for the data-driven mixture prior that includes only trials of DAAs, there is a single mode at HRs between 0.1 and 0.2. Because the PINETREE data were strongly consistent with this prior distribution, the prior served to strengthen the posterior distribution confidence of HR effect estimates, with the narrowest credible interval being that of the DAA mixture prior (HR 0.14; 95% CrI 0.06–0.33).

## Discussion

Bayesian analysis represents an alternative approach to the design and analysis of clinical trials. Bayesian methods provide an intuitive, robust, and methodologically rigorous approach to incorporation of information (e.g., expert opinion, historical data) into the design and analysis of clinical trials. Moreover, these methods make explicit the inherent assumptions encoded in the statistical approach (within the prior and the observed data) and quantify the uncertainty with probability statements [3]. In appropriate settings, this may allow for the reduction in the sample sizes of clinical trials, decreasing costs and trial duration and facilitating recruitment [25,26]. Though these approaches remain rare in

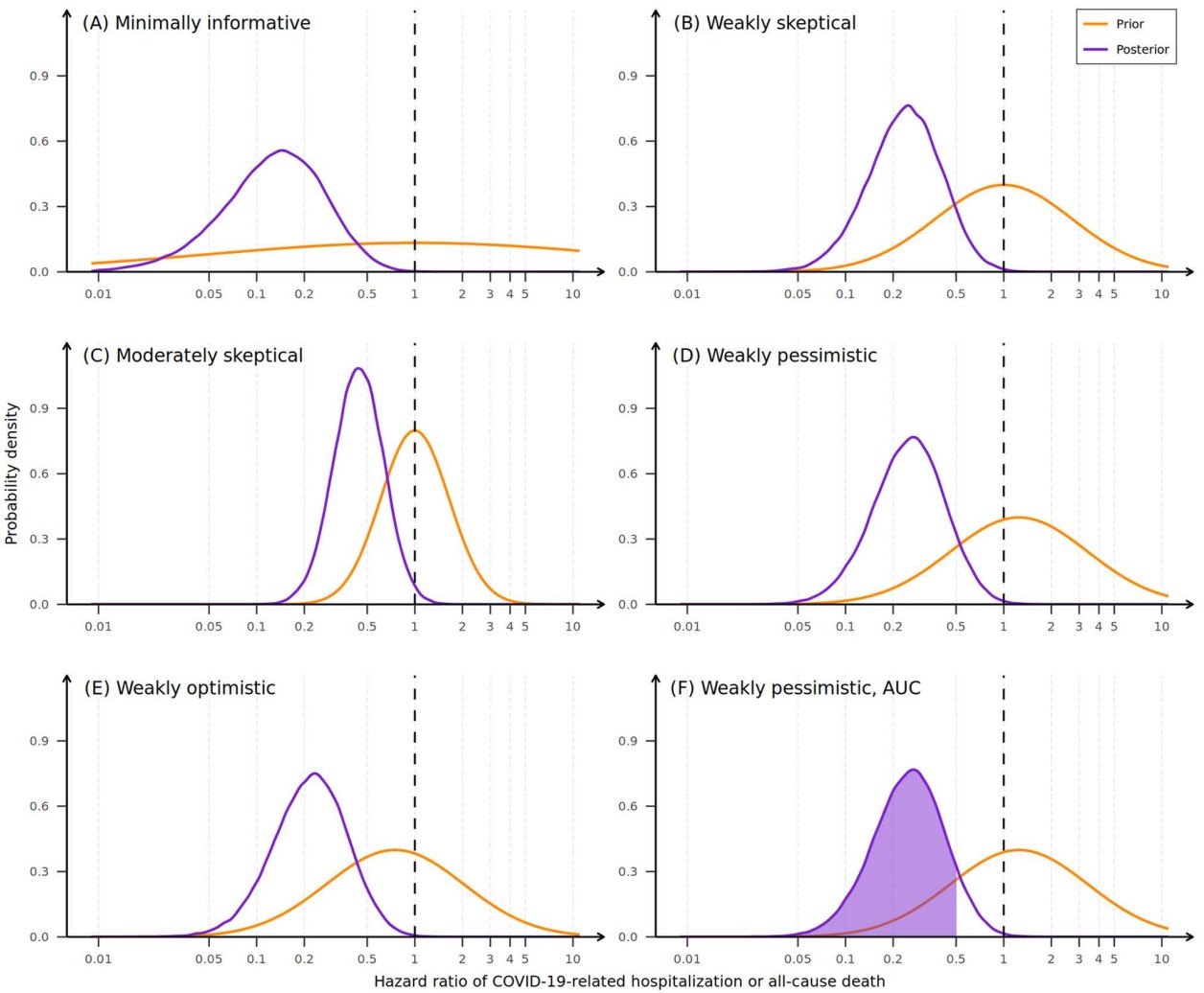

**Fig 3. Prior (orange) and posterior (purple) distributions for the five reference priors for the HR of the treatment effect of RDV on COVID-19–related hospitalization or all-cause death.** HR = hazard ratio; RDV = remdesivir; AUC = area under the curve. Orange lines indicate the reference prior probability distribution. Purple lines indicate the posterior probability distribution. The vertical dotted reference line indicates the no treatment effect threshold (i.e., HR = 1). Probabilities for HRs less than a specific threshold are calculated as the area under the posterior probability curve to the left of that threshold (e.g., panel **F**). Alt text: Graphs depicting posterior distribution, which represents the effects of incorporating the trial data into each of the five reference priors. The final panel shows a shaded region depicting the area under the curve, which represents the posterior probability for the hazard ratio being less than a specific threshold. Reference prior distributions are shown in orange, and the posterior distributions are shown in purple.

regulatory submissions [25], recent precedent has been set for the use of Bayesian statistical approaches to achieve drug approval. In 2018, the United States Food and Drug Administration (U.S. FDA) approved to expand the indication for belimumab—a treatment for systemic lupus erythematosus (SLE)—to include the treatment of children 5–17 years of age utilizing a Bayesian dynamic borrowing approach that incorporated historical adult trial data into a prior distribution used for the analysis [27]. The primary analysis of the randomized, double-blind, placebo-controlled Phase 3 study to evaluate the effectiveness of REBYOTA—a treatment for infection with *Clostridioides difficile*—used a Bayesian model to formally incorporate data from a previous Phase 2 placebo-controlled study, gaining approval in 2022 [28]. In January 2026, the

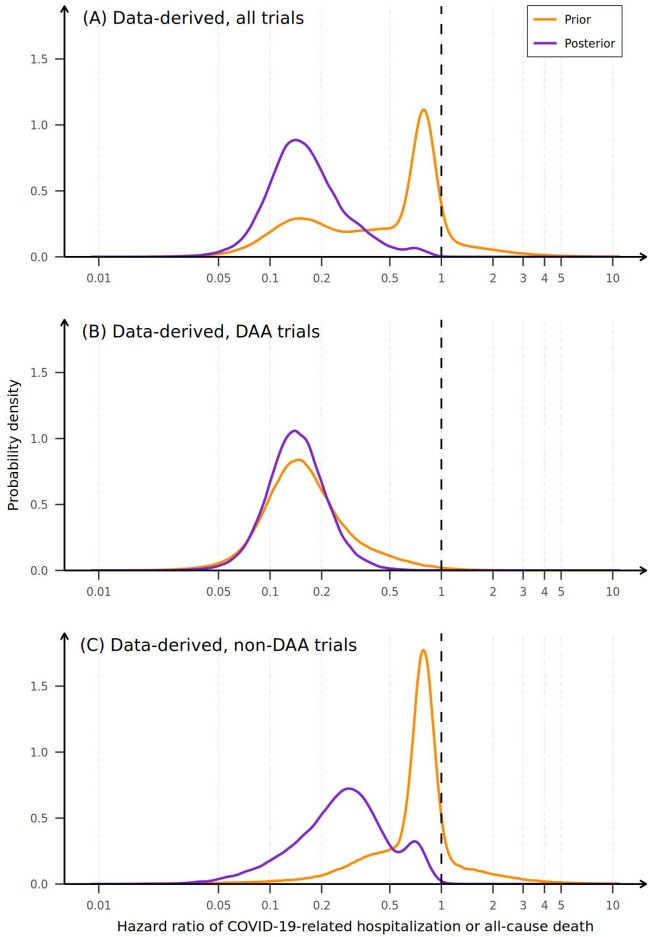

**Fig 4. Prior (orange) and posterior (purple) distributions for the three data-driven priors for the HR of the treatment effect of RDV on COVID-19–related hospitalization or all-cause death.** HR = hazard ratio; RDV = remdesivir; DAA = direct-acting antiviral. Orange lines indicate the data-driven prior probability distribution. Purple lines indicate the posterior probability distribution. The vertical dotted reference line indicates the no treatment effect threshold (i.e., HR = 1). Probabilities for HRs less than a specific threshold are calculated as the area under the posterior probability curve to the left of that threshold (see Fig 3F for example). **(A)** Mixture prior, all trials: the posterior probability reallocates the mode of the prior probability distribution most consistent with the PINETREE data. **(B)** Mixture prior, DAA trials: the variance of the posterior probability density decreases representing increased precision of the effect estimate and a narrower 95% credible interval. **(C)** Mixture prior: non-DAA trials: The posterior probability density is shifted to the right (towards larger HRs) by the large prior probability mass at HRs between 0.5 and 1. Alt text: Graphs depicting posterior distribution, which represents the effects of incorporating the trial data into each of the three data-driven priors. Reference prior distributions are shown in orange, and the posterior distributions are shown in purple.

U.S. FDA published draft guidance for the use of Bayesian methodology in clinical trials of drug and biological products heralding a new era of statistical analysis for clinical trials supporting the effectiveness and safety of drugs [29].

In our analysis, the utilization of historical data from prior RCTs exemplified two fundamental properties of Bayesian data analysis: (1) that "Bayesian inference is reallocation of credibility across possibilities" [30] and (2) that the precision of the point estimate, represented by the width of the posterior credible interval, is increased when data are consistent with prior assumptions. First, in the mixture prior distribution of all 9 RCTs, the probability mass of the bimodal prior distribution was reallocated to those possibilities that were most likely based on the PINETREE trial data (Fig 4A). Second, in the mixture prior of the DAA trials, because the PINETREE trial data were consistent with the mode of this prior, the estimate of the HR is the narrowest of the estimates across all 8 priors (Fig 4B).

There are two other advantages to the Bayesian analytical framework illustrated above. First, the use of a range of prior distributions allows for rigorous mathematical incorporation of a wide spectrum of possible **clinical opinions** regarding the treatment effect of RDV. Despite these differing prior beliefs, we can conclude that there is at least a 0.989 probability of benefit of RDV for the reduction of COVID-19–related hospitalization or all-cause death across all 8 "opinions" encoded by the prior probability distributions. Second, the probabilities for different magnitudes for the treatment effect can be estimated. These probabilities are easy to understand and can be directly incorporated into clinical decision-making. For example, assume a clinician believed that there was only a 15% chance that RDV reduced COVID-19–related hospitalizations or all-cause death by 40% (i.e., P(HR<0.6)=0.15; corresponding to the moderately skeptical prior). Incorporating the new data obtained in the PINETREE trial, she can update her belief and conclude that, based on these new data, there is now an 80% chance of at least that same treatment effect (i.e., P(HR<0.6)=0.8). She may then use this new probability on the treatment effect to consider whether these data warrant a change to her clinical practice. Another clinician may instead be inclined to weigh prior RCTs into his decision but may still have some skepticism about large treatment effects. He may believe that there is approximately a 40% chance that the HR is between 0.75 and 1, similar to that of the mixture prior of all 9 historical RCTs. With the new PINETREE data, he can update his belief concluding that there is actually a 70% chance that the HR is 0.2 or less.

## Limitations

Researchers must exercise caution when utilizing Bayesian methodological frameworks. The prior distributions used for Bayesian inference influence the resulting posterior effect estimates, especially in experiments with smaller sample sizes [31,32]. Therefore, the prior distributions need to be selected carefully. Using a prior that does not contain any useful information about the parameters to be estimated (e.g., a diffuse prior probability distribution that allocates probability density to clinically impossible HRs such as below 0.0001 or above 1000) would ignore information and render the model output more challenging to interpret.

Similarly, prior distributions dependent on historical RCTs may be considered inconsistent with the methodology of the new trial data. In this study, because of the limited number of RCTs with similar trial and statistical methodologies as those used in PINETREE, studies that reported hazard ratios, risk ratios, and odds ratios for study endpoints that included hospitalization or death were incorporated in the data-driven prior distributions. Selecting historical studies with similar trial population, trial methodology, clinically compatible outcome measures, and statistical methodologies that approximate the current trial is recommended and may mitigate some of this concern. Several other mitigation strategies have been employed in the literature to address lack of compatibility, with the most common being down-weighting historical trial data (e.g., mixing the prior distribution with a minimally informative prior) to reduce their influence on the calculated posterior [33].

Further, interpretation and communication of results presented under a variety of different prior distributions can be challenging. Researchers advocate that several prior distributions be used to ensure that the clinical trial results are robust to prior selection and represent the diversity of clinical opinion. With several posterior distributions generating several posterior point estimates and credible intervals, authors must report these results with care. In practice, one prior may be selected as the primary prior to be used in the analysis while several other priors be labeled as sensitivity analyses. If results are not robust to prior selection, researchers should report this, explain what those priors represent, and discuss the resulting implications for clinical decision-making.

Lastly, this was an unplanned Bayesian reanalysis of the primary endpoint. Frequentist analyses that are not prespecified are vulnerable to increased error rates (especially type I error) because of multiple hypothesis testing [34]. Importantly, we assessed the prespecified primary endpoint of the PINETREE trial without modification. Moreover, unlike in frequentist analysis, error rates in Bayesian statistics vary by prior distribution and repeated estimation does not modify the prior distribution but rather adds information via the likelihood to the posterior distribution. This difference between

frameworks is taken advantage of in strict Bayesian approaches to interim analyses and in Bayesian adaptive trial designs where the emphasis is placed on estimation of the posterior probabilities of treatment effects [35].

### Future research

There were several secondary endpoints in PINETREE including the composite of COVID-19–related medically attended visits or death from any cause by day 28 and the time to alleviation of baseline COVID-19 symptoms based on FLU-PRO Plus questionnaire. These data could also be reanalyzed using a Bayesian approach using Bayesian empirical likelihood in order to gain similar insights to our analysis of the primary endpoint.

### Conclusions

In a Bayesian reanalysis of the PINETREE trial, there was a 0.989 or greater probability that treatment with RDV reduced the risk of the primary outcome of COVID-19–related hospitalization or all-cause death across all prior probability distributions. This analysis also elucidated the posterior probability of the treatment effect under a wide range of prior beliefs demonstrating the robustness of the trial results to different prior distributions, and the flexibility, intuitiveness, and informativeness that a Bayesian analysis of a clinical trial affords.

### Supporting information

**S1 Table. Characteristics of the randomized controlled trials selected for calculation of the data-driven mixture priors.**
(DOCX)

**S1 Appendix. PINETREE trial protocol and statistical analysis plan.**
(PDF)

**S2 Appendix. Mathematical description of the statistical modeling.** The Cox proportional hazards model, the Bayesian model including development of the reference and data drive priors, and the Monte Carlo estimation of the parameters of interest are described.
(PDF)

**S3 Appendix. CONSORT checklist.**
(DOCX)

**S4 Appendix. Related manuscript.** The New England Journal of Medicine publication of the PINETREE trial titled "Early Remdesivir to Prevent Progression to Severe Covid-19 in Outpatients" (reference 9).
(PDF)

**S5 Appendix. Related manuscript appendix.** The appendix accompanying The New England Journal of Medicine publication of the PINETREE trial titled "Early Remdesivir to Prevent Progression to Severe Covid-19 in Outpatients" (reference 9).
(PDF)

**S6 Appendix. Data sharing statement accompanying PINETREE trial publication.**
(PDF)

### Author contributions

**Conceptualization:** Mazin Abdelghany, Fang Yu, Stephen Rennard, Yeongjin Gwon.

**Formal analysis:** Mazin Abdelghany.

**Methodology:** Mazin Abdelghany, Fang Yu, Stephen Rennard, Yeongjin Gwon.

**Software:** Mazin Abdelghany.

**Supervision:** Yeongjin Gwon.

**Visualization:** Mazin Abdelghany.

**Writing – original draft:** Mazin Abdelghany.

**Writing – review & editing:** Mazin Abdelghany, Fang Yu, Stephen Rennard, Yeongjin Gwon.

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
