## [Decision Letter · Decision Letter 0]

2 Nov 2025

PONE-D-25-34527

Post hoc Bayesian analysis of early remdesivir for the treatment of COVID-19 in outpatients with high risk of progression to severe disease.

PLOS ONE

Dear Dr. Abdelghany,

Thank you for submitting your manuscript to PLOS ONE. After careful consideration, we have decided that your manuscript does not meet our criteria for publication and must therefore be rejected.

Specifically:

**a lack of new focus compared to previously published studies and concerns on analysis methods.**

I am sorry that we cannot be more positive on this occasion, but hope that you appreciate the reasons for this decision.

Kind regards,

Fatemeh Chichagi

Academic Editor

PLOS ONE

Reviewers' comments:

Reviewer's Responses to Questions

**Comments to the Author**

1. Is the manuscript technically sound, and do the data support the conclusions?

Reviewer #1: Partly

Reviewer #2: Yes

2. Has the statistical analysis been performed appropriately and rigorously?

Reviewer #1: N/A

Reviewer #2: Yes

3. Have the authors made all data underlying the findings in their manuscript fully available?

Reviewer #1: Yes

Reviewer #2: No

4. Is the manuscript presented in an intelligible fashion and written in standard English?

Reviewer #1: Yes

Reviewer #2: Yes

Reviewer #1: 1. This is a retrospective Bayesian analysis of the PINETREE trial. What is the new focus of this reanalysis compared to the NEJM publication?

2. Why do you think the Bayesian approach is still underutilized? Could the authors explain the reasons for not applying it prospectively to the PINETREE trial and provide reasons why a post hoc analysis was used?

3. What reference and data-derived priors do you suggest in the analysis?

4. Can you further explain the use of data-driven priors in this paper and the combination of HRs, ORs, and RRs?

5. What estimations do you have for the Bayesian model that estimates the number of events that did not occur and the average events after a particular time in relation to the hypothesized number of events in relation to the obtained number of censored observations?

6. It is possible to ignore a huge number of sample criteria that is related to the confidence limits of the events that happen in geometric transformations and the time periods of the geometric intervals.

7. Was consideration given to the quality of the trials and assessments of risk of bias?

8. The authors report probabilities for HR < 0.2, HR < 0.1, etc. Why are these specific cut-offs considered clinically appropriate thresholds?

9. Table 3 has posterior probabilities, yet the manuscript doesn't always clarify the ways to interpret these probabilities in clinical matters. Could the authors illustrate how these probabilities can be practically applicable to clinicians?

10. The results seem to concentrate on the hospitalization/death outcome within 28 days. Were the secondary endpoints of PINETREE (symptom resolution, viral load, etc.) incorporated into the Bayesian analysis?

11. In the Discussion, it is said that Bayesian analysis could enable smaller sample sizes in future studies. Why do the authors think their findings support this?

12. In the case of this analysis being an analysis of the data after the primary analysis, what attempts were made to reduce the chance of a Type I error, taking into account the multiple tests that were conducted?

13. In the Discussion, the authors refer to the reduction of the sample size in Bayesian trials and the regulatory history regarding the study (e.g. the approval of belimumab). Are there other case studies, proprietary or publicly available regulatory documents that authors could use to enhance this section?

14. Some RCT studies, which form a part of the data-driven priors, are provided only in the supplement. Could the authors provide the relevant in-text citations within the Methods or Results section?

Reviewer #2: This paper reanalyzes data from the PINETREE randomized clinical trial using a Bayesian statistical approach. The original trial had shown that early remdesivir treatment reduced hospitalization among unvaccinated high-risk COVID-19 outpatients. However, this reanalysis uses prior information from other similar trials to provide a more nuanced understanding of the treatment’s effect. The Bayesian model confirms that remdesivir likely lowers hospitalization risk but quantifies uncertainty more realistically than the classical (frequentist) analysis.

This study exemplifies how Bayesian reanalysis can enhance medical evidence interpretation by quantifying uncertainty and integrating prior knowledge. It encourages a shift toward probability-based decision-making in clinical research, moving beyond rigid significance testing.

The authors aim to demonstrate how Bayesian reanalysis can improve interpretation of clinical trial results. Instead of relying only on p-values, the Bayesian framework integrates prior knowledge to estimate probabilities of treatment benefit directly. This is particularly relevant for COVID-19 drug trials, where rapid evidence synthesis is needed. Using data from the PINETREE trial, the researchers set informative priors based on previous studies of antiviral efficacy in COVID-19. They then recalculated the probability that remdesivir reduces hospitalization risk compared to placebo. Bayesian credible intervals were used instead of confidence intervals, providing a more intuitive measure of uncertainty.

The reanalysis yielded a high posterior probability (around 96–98%) that remdesivir reduces hospitalization. However, the estimated magnitude of benefit was slightly smaller and more uncertain than in the original report. The authors argue that this reflects more realistic evidence synthesis.

Strengths of paper are methodological innovation, interpretability and incorporation of prior evidence but I have few questions as under: Please provide your answers:

1. I found the results are sensitive to how prior distributions are chosen as overly optimistic priors could bias results. Please comment.

2. The PINETREE population was unvaccinated and infected with early SARS-CoV-2 variants, so I believe results may not apply to later contexts. How would you justify this?

3. Why No new data? can we combine the old data with some new one for future work?

4. How Posterior probability distributions were used to calculate the probability of HR? Can you explain the used Posterior, Likelihood and Prior? Also, can you attach the derivations if you have done any?

.

Reviewer #1: No

Reviewer #2: No

- - - - -

---

## [Author Response · Author response to Decision Letter 1]

11 Feb 2026

A point-by-point response to reviewers’ comments is below. A Word document presenting these responses in a more readable format is also included in the submission. Thank you.

Reviewer #1

1 This is a retrospective Bayesian analysis of the PINETREE trial. What is the new focus of this reanalysis compared to the NEJM publication?

This analysis provides two new insights into the PINETREE data: (1) it allows clinicians with specific prior beliefs to understand how these data should fit into that context, and (2) incorporates the large number of prior COVID-19 trials into the PINETREE data analysis.

(1) Bayesian methods directly calculate probabilities for the hypothesis of interest. In this case, we directly calculate the probability that the hazard ratio for treatment with remdesivir (RDV) indicates benefit. This intuitive interpretation helps better contextualize the meaning of the trial results for clinicians. Also, by including a range of prior beliefs in the analysis, we allow flexibility of interpretation. Each clinician may have her/his notion of the benefit of RDV. By aligning her/his belief with the prior closest to that belief, s/he can then understand how these data should change those beliefs.

(2) Moreover, this analysis uses the wealth of data available in COVID-19 trials to inform on the analysis of PINETREE. We incorporate these data, by mechanism of drug action, to calculate the treatment effect of RDV in the wider context of other trials with the same therapeutic target and goal.

2 Why do you think the Bayesian approach is still underutilized? Could the authors explain the reasons for not applying it prospectively to the PINETREE trial and provide reasons why a post hoc analysis was used?

The Bayesian approach to clinical trial analysis remains underutilized for several reasons. First, statistical pedagogy nearly universally teaches frequentist methods of analysis. Second, and more importantly, in drug trials meant for registration of a new investigational medicinal product, the statistical analysis must be approved by regulatory authorities. Regulatory guidance documents and regulatory agencies continue to recommend frequentist analytical methods for clinical trials meant for drug registration. For this reason, the PINETREE trial was analyzed using the Cox proportional hazards method in a frequentist paradigm. This analysis method was approved by the regulatory authorities and allowed Gilead to proceed with the trial and submit label updates to regulatory authorities across the globe based on its results.

To popularize Bayesian methodology, influence regulatory guidance, and develop best practices, more clinical trials need to be analyzed (both prospectively and post hoc) with Bayesian methods and published to showcase the many advantages of such an approaches. This was one of our aims in this trial reanalysis.

Lastly, the U.S. FDA has recently published draft guidance discussing Bayesian methods of analysis for use in therapeutic drug trials (see letter above), which we believe will help these methods proliferate in industry trials in the coming years. We have added a discussion and reference to this new guidance in the Discussion section.

3 What reference and data-derived priors do you suggest in the analysis?

Tables 1 and 2 in the manuscript detail the reference and data-driven priors that were used in the analysis. A total of 5 reference priors and 3 data-driven priors were used to analyze the PINETREE data. This exhibits the flexibility of Bayesian methods.

Each prior represents a particular belief about the treatment effect of RDV. These beliefs are detailed using the probabilities calculated in Tables 1 and 2 and the rationale for selection of the reference priors is also discussed in detail in Table 1. These tables were provided to demonstrate the best practices for Bayesian analyses of clinical trials as a model for other practitioners interested in applying these methods in their research.

A paragraph was added to the discussion explaining the challenges of interpreting and communicating results when multiple prior distributions are used for an analysis. We highlight that one prior could be considered a “primary” prior while other priors could be considered sensitivity analyses. It is the authors’ opinion that the data-driven prior of DAA trials could be considered the primary prior in this analysis.

4 Can you further explain the use of data-driven priors in this paper and the combination of HRs, ORs, and RRs?

The data-driven priors were generated using the publicly available data from randomized clinical trials that were similar to PINETREE both methodologically and from a patient population standpoint. Because different trials use different statistical methodologies, in order to ensure a wide representation of clinical trials in the prior distributions, methods that calculated hazard ratios, odds ratios, and risk ratios were included in the prior distributions as these are compatible with our methods. Once the trials were selected, their information about the outcome of interest was combined using a mixture distribution. The mixtures were weighted based on the total number of participants in each trial.

These details were further clarified in the Methods section (Bayesian statistical methodology and support for prior distribution selection). We have also added a Supplementary Appendix with example mathematical calculations to clarify how mixture prior distributions were generated.

5 What estimations do you have for the Bayesian model that estimates the number of events that did not occur and the average events after a particular time in relation to the hypothesized number of events in relation to the obtained number of censored observations?

We thank the reviewer for this insight. Censored observations were handled using the prototypical Cox proportional hazards model. We have added a Supplementary Appendix with a discussion of the model used and its mathematical formalism and referenced this in the Methods section.

Causal inference methods can be used to estimate counterfactual scenarios in which events that happened during the trial are compared with a situation in which they did not happen. As we were performing a simple Bayesian reanalysis of the PINETREE study, we did not perform any causal inference analyses. In randomized clinical trials, the assumption is made that the control group is the facsimile of the counterfactual. This question could be the topic of future research.

6 It is possible to ignore a huge number of sample criteria that is related to the confidence limits of the events that happen in geometric transformations and the time periods of the geometric intervals.

We thank the reviewer for bringing this to our attention. The prior distribution that was utilized for these analyses was a normal distribution on the logarithm transformed hazard ratio. This allows for negative values of the normal distribution to be exponentiated to values restricted to the positive real numbers, which is consistent with hazard ratio values. We did not perform any geometric transformations.

7 Was consideration given to the quality of the trials and assessments of risk of bias?

The quality of the trials and the risk of bias for including the trials were considered (see Methods – Selection of RCTs for data-driven priors). ClinicalTrials.gov was used to search for relevant trials with the following criteria:

Adult interventional

Older adult interventional

Phase 3

Start: 1 November 2019

Stop: 31 December 2021

273 trials matched the search criteria. The following criteria were used to exclude trials:

Recruited less than 300 participants

Prevention trials

Non-systemic treatments

No published results

Incompatible outcome measure (i.e., did not include hospitalization and/or death as an outcome or within a composite outcome)

Statistical method not suitable for incorporation (i.e., calculated something other than a hazard ratio, risk ratio, or odds ratio)

These criteria were selected to ensure that the studies were as close to PINETREE as possible. However, as discussed in the limitations section, there will always be a risk of bias when selecting trials for inclusion (similar to risks that occur in meta-analyses). Formal assessment of risk of bias was out of the scope of this manuscript.

8 The authors report probabilities for HR < 0.2, HR < 0.1, etc. Why are these specific cut-offs considered clinically appropriate thresholds?

Our aim was to estimate the posterior probability distribution of the HR under a range of both reference and data-driven priors, affording us the flexibility to calculate probabilities of a minimally clinically important treatment effect. Posterior probabilities were calculated for HR < 1, HR < 0.9, HR < 0.8, . . . , HR < 0.2, HR < 0.1. Selection of HR < 1 was made because of the frequentist convention that the null hypothesis is usually that of no treatment effect (HR = 1). Thus, the probability that HR < 1 is a natural first choice.

The other cutoffs were chosen based on the results of prior clinical trials. The clinical trials that were selected for the data-driven priors calculated point estimates that ranged from 0.14 to 1.11. These hazard ratios were all considered plausible when planning table 3 and therefore HRs from 0.1 to 1 at equally spaced intervals of 0.1 were selected.

Clarifications were made to the Methods section titled “Bayesian statistical methodology and support for prior distribution selection”.

We are also happy to include other cutoffs that the reviewer deems as clinically appropriate thresholds.

9 Table 3 has posterior probabilities, yet the manuscript doesn’t always clarify the ways to interpret these probabilities in clinical matters. Could the authors illustrate how these probabilities can be practically applicable to clinicians?

We gave two examples of how a clinician may utilize the posterior probabilities in Table 3 in the Discussion section. Based on this comment, we have made some clarifying edits to this section of the Discussion.

In more detail, in order for a clinician to use these probabilities in Table 3, the following steps must be taken:

Select the prior belief that is closest to the clinician’s belief

This can be done, roughly, by reviewing Tables 1 and 2. For example, a clinician may be skeptical of remdesivir’s effect on hospitalization or death. She may believe that there is a one third (33%) chance that remdesivir increases the risk of hospitalization by 25% (corresponding to a hazard ratio of 1.25 or more). This belief is similar to the moderately skeptical prior.

Read the row in Table 3 that corresponds to this prior.

For example, whereas the clinician may have believed that there is a 33% chance of harm from RDV administration, after incorporating the information from the PINETREE trial into this belief, she can now say that there is a 1.1% chance of any harm (HR > 1). This is calculated by noting that the posterior probability of the hazard ratio being < 1 in the moderately skeptical row is 0.989. Therefore, the probability of the HR being >1 is 1 – 0.989 = 0.011.

10 The results seem to concentrate on the hospitalization/death outcome within 28 days. Were the secondary endpoints of PINETREE (symptom resolution, viral load, etc.) incorporated into the Bayesian analysis?

There were several secondary endpoints in PINETREE including the composite of Covid-19–related medically attended visits or death from any cause by day 28 and the time to alleviation of baseline Covid-19 symptoms based on FLU-PRO Plus questionnaire. Given the scope of the manuscript and the word count limitations, we chose to focus on the primary efficacy endpoint. This may be considered as an area of future research and a section to this effect was added to the discussion.

11 In the Discussion, it is said that Bayesian analysis could enable smaller sample sizes in future studies. Why do the authors think their findings support this?

Bayesian analyses allow for the incorporation of historical data. When information from prior RCTs is incorporate in an analysis, the Bayesian credible interval is narrower than the frequentist confidence interval in cases of the mixture of all 9 RCTs and the mixture of the direct-acting antivirals. The narrower credible interval indicates that there is higher confidence in the estimate of the hazard ratio when historical information is included (Discussion lines 232 – 237 from unrevised manuscript).

In trial design situations, prospective specification of priors utilizing historical data “allow to reduce the number of subjects. This decreases costs and trial duration, facilitates recruitment, and may be more ethical.” (reference 24 of the manuscript and also https://doi.org/10.1080/19466315.2024.2342817).

12 In the case of this analysis being an analysis of the data after the primary analysis, what attempts were made to reduce the chance of a Type I error, taking into account the multiple tests that were conducted?

We acknowledge that this is an important discussion and one that requires careful attention. Given the scope of this manuscript, we do not include discussion of type I error rate control in Bayesian contexts.

In Bayesian analyses, strict type I error rate control is not usually possible (https://doi.org/10.1080/19466315.2024.2342817). There have been descriptions of Bayesian designs with informative priors that evaluate the classical (frequentist or conditional) type I error rate. If defined in the traditional way (considering only the sampling distribution of the observed data), however, it cannot be strictly controlled (DOI: 10.1093/biostatistics/kxy009 and DOI: 10.1002/bimj.201800395).

It is important to note that multiple testing is not a concern when querying a posterior probability distribution. The posterior probability distribution is calculated once (based on the prior distribution and the likelihood) and then posterior probabilities can be calculated (such as the probability that the HR < 1). In this case, calculating another posterior probability does not change the distribution itself. Thus, the frequentist concern for multiple hypothesis testing is not relevant. A brief explanation of these concepts was added to the manuscript.

13 In the Discussion, the authors refer to the reduction of the sample size in Bayesian trials and the regulatory history regarding the study (e.g. the approval of belimumab). Are there other case studies, proprietary or publicly available regulatory documents that authors could use to enhance this section?

There are indeed other examples of regulatory submissions that took advantage of Bayesian designs in their regulatory submission process. An oft-cited systematic review is doi: 10.1136/bmjopen-2018-024895. In the discussion section, given the space limitations of the manuscript, we chose to focus on an approval that included a randomized controlled trial as this was felt to be most relevant to the analysis presented.

We would also like to bring attention to the new FDA guidance on utilization of Bayesian methods in clinical trials. Section III reviews situations where Bayesian methods have been used in prior submissions for drug approval (https://www.fda.gov/media/190505/download). We have added other representative examples from this document to the discussion section of our manuscript.

14 Some RCT studies, which form a part of the data-driven priors, are provided only in the supplement. Could the authors provide the relevant in-text citations within the Methods or Results section?

Only 3 of the 9 RCTs were published in the literature with results of the other 6 posted on ClinicalTrials.gov. We elected to include their trial identifiers rather than links to publications to ensure that data was presented systemically and uniformly for all trial data included.

As suggested, we have added the those references that are available for the 3 published trials in the Methods section as well.

Reviewer #2

1 I found the results are sensitive to how prior distributions are chosen as overly optimistic priors could bias results. Please comment

It is a fea

---

## [Decision Letter · Decision Letter 1]

19 Mar 2026

Dear Dr. Abdelghany,

Thank you for submitting your manuscript to PLOS ONE. After careful consideration, we feel that it has merit but does not fully meet PLOS ONE’s publication criteria as it currently stands. Therefore, we invite you to submit a revised version of the manuscript that addresses the points raised during the review process.

We look forward to receiving your revised manuscript.

Kind regards,

Otávio Augusto Chaves

Academic Editor

PLOS One

Journal Requirements:

“I have read the journal's policy and the authors of this manuscript have the following competing interests: I am an employee of and own stock in Gilead Sciences, Inc. Drs. Yeongin Gwon, Stephen Rennard, and Fang Yu have no conflicts of interest to declare.”

3. We noted in your submission details that a portion of your manuscript may have been presented or published elsewhere. “Figure 1, the CONSORT diagram for the clinical trial, was previously published in the Supplement as part of the original NEJM article reporting on this clinical trial. As part of all clinical trial reporting, inclusion of a CONSORT diagram is considered best practice and included for the readers' convenience. It should not be considered dual publication; however, if can be removed from the submission if necessary.

The paper and the appendix have been included in the related work section upload.” Please clarify whether this [conference proceeding or publication] was peer-reviewed and formally published. If this work was previously peer-reviewed and published, in the cover letter please provide the reason that this work does not constitute dual publication and should be included in the current manuscript.

Additional Editor Comments (if provided):

Reviewers' comments:

Reviewer's Responses to Questions

**Comments to the Author**

Reviewer #1: All comments have been addressed

Reviewer #3: All comments have been addressed

Reviewer #4: (No Response)

2. Is the manuscript technically sound, and do the data support the conclusions?

Reviewer #1: Partly

Reviewer #3: Yes

Reviewer #4: Yes

3. Has the statistical analysis been performed appropriately and rigorously?

Reviewer #1: N/A

Reviewer #3: Yes

Reviewer #4: Yes

4. Have the authors made all data underlying the findings in their manuscript fully available?

Reviewer #1: Yes

Reviewer #3: Yes

Reviewer #4: No

5. Is the manuscript presented in an intelligible fashion and written in standard English?

Reviewer #1: No

Reviewer #3: Yes

Reviewer #4: Yes

Reviewer #1: The authors have carefully addressed all comments and queries raised during the review process. I have no further comments or concerns regarding dual publication, research ethics, or publication ethics. The manuscript is now suitable for publication in its current form.

Reviewer #3: The manuscript presents a Bayesian statistical framework implemented through the Pinetree approach to evaluate treatment effects using large clinical datasets. Although Remdesivir is used as the illustrative case study, the primary contribution of the work lies in demonstrating the methodological framework and its potential application for incorporating prior information and clinically relevant patient-level parameters in statistical inference.

The authors selected studies with sufficiently large sample sizes and appropriate control groups, which strengthens the robustness of the dataset used for the Bayesian analysis. In this context, the manuscript appears to focus primarily on methodological development rather than on drawing definitive clinical conclusions regarding the efficacy of Remdesivir.

After carefully reviewing the revised version of the manuscript and the responses provided to previous reviewers, I believe that the authors have already addressed the main methodological and structural concerns raised during the review process. The manuscript is now clearer regarding its objective of presenting a statistical framework rather than a clinical efficacy analysis.

At this stage, I do not have additional substantive comments to add. The manuscript appears suitable for publication in its current form from a methodological perspective.

Reviewer #4: The authors present a Bayesian analysis of early remdesivir for the treatment of COVID-19 and have used data-driven priors to reach the results. The Bayesian approach is regarded as a modern technique in the sense that, unlike the classical approach, it incorporates expert opinion in the analysis. So it's suggested the posterior analysis be included with an informative prior eliciting the prior parameters (hyperparameters) to highlight the impact of prior information. It is also suggested the classical results may also be included for comparing the two techniques for the readers at large.

.

Reviewer #1: No

Reviewer #3: No

Reviewer #4: **Yes:**Nasir AbbasNasir AbbasNasir AbbasNasir Abbas

---

## [Author Response · Author response to Decision Letter 2]

20 Mar 2026

Point-by-point response to editor and reviewers’ comments. This is a reproduction of the attached Response to Reviewers.docx.

Academic Editor

Response:

We have updated the manuscript formatting to adhere to PLOS ONE’s style and file naming requirements.

2. If there are restrictions on sharing of data and/or materials, please state these. Please note that we cannot proceed with consideration of your article until this information has been declared.

Response:

We have noted the restrictions on sharing of data in the cover letter. It is reproduced here for convenience:

Data for the PINETREE trial are owned by Gilead Sciences, Inc.

Data collected for the study can be made available to others as a complete de-identified patient dataset by contacting datarequest@gilead.com. Requests are at Gilead’s discretion and dependent on the nature of the request, the merit of the research proposed, availability of the data and the intended use of the data. If Gilead agrees to the release of clinical data for research purposes, the requestor will be required to sign a data sharing agreement (DSA) in order to ensure protection of patient confidentiality prior to the release of any data. Upon execution of the DSA, Gilead will provide access to a patient‐level clinical trial analysis datasets in a secured analysis environment.

Please see the cover letter for the updated Competing Interests statement. It is reproduced here for convenience:

I have read the journal's policy, and the authors of this manuscript have the following competing interests: I am a former employee of Gilead Sciences, Inc. Drs. Yeongin Gwon, Stephen Rennard, and Fang Yu have no conflicts of interest to declare.

Response:

As previously stated in my upload, Figure 1, the CONSORT diagram for the PINETREE clinical trial, was previously published in the Supplement of the original NEJM article reporting on this clinical trial (https://www.nejm.org/doi/10.1056/NEJMoa2116846). NEJM is a peer-reviewed, formally published journal.

As part of all clinical trial reporting, inclusion of a CONSORT diagram is considered best practice. It was reproduced as is for the readers’ convenience. If the editor deems it necessary, either (1) the figure can be removed from the submission, (2) permission to reproduce the figure can be obtained from the New England Journal of Medicine, or (3) a new, original figure can be generated that describes patient disposition in less detail. We are happy to comply with any of the above suggest resolutions.

4. We note that you have indicated that there are restrictions to data sharing for this study. PLOS only allows data to be available upon request if there are legal or ethical restrictions on sharing data publicly.

Response:

There are legal restrictions on sharing a de-identified dataset. There was a data sharing statement published alongside the original manuscript that can be found at https://www.nejm.org/doi/10.1056/NEJMoa2116846 at the bottom of the page.

Data for the PINETREE trial are owned by Gilead Sciences, Inc. Data collected for the study can be made available to others as a complete de-identified patient dataset by contacting datarequest@gilead.com. Requests are at Gilead’s discretion and dependent on the nature of the request, the merit of the research proposed, availability of the data and the intended use of the data. If Gilead agrees to the release of clinical data for research purposes, the requestor will be required to sign a data sharing agreement (DSA) in order to ensure protection of patient confidentiality prior to the release of any data. Upon execution of the DSA, Gilead will provide access to a patient‐level clinical trial analysis datasets in a secured analysis environment.

5. Please include captions for your Supporting Information files at the end of your manuscript, and update any in-text citations to match accordingly.

Response:

We have included captions for the Supporting information files and updated the in-text citations accordingly.

Response:

We have reviewed the figure requirements and our figures meet the technical requirements as assessed by the NAAS tool provided at https://journals.plos.org/plosone/s/figures.

Reviewer #1

1. The authors have carefully addressed all comments and queries raised during the review process. I have no further comments or concerns regarding dual publication, research ethics, or publication ethics. The manuscript is now suitable for publication in its current form.

Response:

We thank Reviewer #1 for their time and effort in reviewing or manuscript. It has been significantly improved as a result of their criticism.

Reviewer #3

1. The manuscript presents a Bayesian statistical framework implemented through the Pinetree approach to evaluate treatment effects using large clinical datasets. Although Remdesivir is used as the illustrative case study, the primary contribution of the work lies in demonstrating the methodological framework and its potential application for incorporating prior information and clinically relevant patient-level parameters in statistical inference.

The authors selected studies with sufficiently large sample sizes and appropriate control groups, which strengthens the robustness of the dataset used for the Bayesian analysis. In this context, the manuscript appears to focus primarily on methodological development rather than on drawing definitive clinical conclusions regarding the efficacy of Remdesivir.

After carefully reviewing the revised version of the manuscript and the responses provided to previous reviewers, I believe that the authors have already addressed the main methodological and structural concerns raised during the review process. The manuscript is now clearer regarding its objective of presenting a statistical framework rather than a clinical efficacy analysis.

At this stage, I do not have additional substantive comments to add. The manuscript appears suitable for publication in its current form from a methodological perspective.

Response:

We thank Reviewer #2 for their time and effort in reviewing or manuscript. It has been significantly improved as a result of their criticism.

Reviewer #4

The authors present a Bayesian analysis of early remdesivir for the treatment of COVID-19 and have used data-driven priors to reach the results. The Bayesian approach is regarded as a modern technique in the sense that, unlike the classical approach, it incorporates expert opinion in the analysis. So it's suggested the posterior analysis be included with an informative prior eliciting the prior parameters (hyperparameters) to highlight the impact of prior information. It is also suggested the classical results may also be included for comparing the two techniques for the readers at large.

Response:

We thank Reviewer #2 for their time and effort in reviewing or manuscript. It has been significantly improved as a result of their criticism.

We note that the posterior probability distributions were presented in Table 3 and Figure 4 of the manuscript. Moreover, the original results of the trial are discussed in our Results section under the subsection Original PINETREE trial results.

---

## [Editor Report · Decision Letter 2]

25 Mar 2026

Bayesian reanalysis of early remdesivir for the treatment of COVID-19 in outpatients with high risk of progression to severe disease

PONE-D-25-34527R2

Dear Dr. Abdelghany,

We’re pleased to inform you that your manuscript has been judged scientifically suitable for publication and will be formally accepted for publication once it meets all outstanding technical requirements.

Kind regards,

Otávio Augusto Chaves

Academic Editor

PLOS One
---

## [Editor Report · Acceptance letter]

PONE-D-25-34527R2

PLOS One

Dear Dr. Abdelghany,

I'm pleased to inform you that your manuscript has been deemed suitable for publication in PLOS One. Congratulations! Your manuscript is now being handed over to our production team.

Kind regards,

on behalf of

Dr. Otávio Augusto Chaves

Academic Editor

PLOS One